# Correlations between Dementia and Loneliness

**DOI:** 10.3390/ijms25010271

**Published:** 2023-12-24

**Authors:** Julia Karska, Magdalena Pszczołowska, Anna Gładka, Jerzy Leszek

**Affiliations:** 1Department of Psychiatry, Wrocław Medical University, Pasteura 10, 50-367 Wrocław, Poland; julia.karska@student.umw.edu.pl (J.K.);; 2Faculty of Medicine, Wrocław Medical University, Pasteura 1, 50-367 Wrocław, Poland

**Keywords:** loneliness, dementia, Alzheimer’s disease

## Abstract

This review describes associations between dementia and loneliness on the neurobiological and epidemiological levels according to the recent body of literature. The aim of this study was to highlight major lines of research in this field. Sociocognitive skills and social interactions present complex interdependencies with dementia which may be explained by two theories. According to the first one, not sufficiently engaging in social or cognitive activities results in brain atrophy. The second one claims that brain neurogenesis and synaptic density are being increased by social connections. The relationship between loneliness and dementia could be mediated by sensory loss, including hearing and visual impairment, as well as depression and psychotic symptoms. Loneliness itself might cause a depletion in sensory and cognitive stimulation which results in a decrease in neural reserve. Certain changes in the structures of the brain caused by loneliness were found in imaging examination. Loneliness appears to be a crucial risk factor for dementia in recent times due to the modern lifestyle and consequences of the outbreak of COVID-19. Additional studies are required to understand more completely the key tenets of this topic and therefore to improve the prevention and treatment of dementia.

## 1. Introduction

Dementia represents a pressing medical and social concern for both the present and the forthcoming years. Around 55 million people worldwide suffer from this condition, and this is expected to rise to 78 million by 2030 and 139 million by the WHO [1]. It also has significant economic implications with global societal costs of dementia of USD 1.3 trillion and with an expected rise to USD 2.8 trillion by 2030 according to the WHO [1]. There are multiple risk factors for this disorder, but one of them occurs to be especially important due to today’s lifestyles—loneliness. This is another rising problem nowadays, affecting in certain countries up to 1 in 3 older people according to the WHO [2]. The crucial influence of social isolation and loneliness on mental and physical health in old age is evident. These factors are linked to psychiatric disorders, including anxiety, depression, psychotic disorders, and Alzheimer’s disease, as well as physical conditions such as cardiovascular diseases and autoimmune diseases [3]. Loneliness can concern the emotional and/or social sphere [4]. Emotional loneliness refers to the lack of an attachment figure (together with feelings of isolation) and includes feeling lonely when not being alone as well. Social loneliness is defined as the absence of a social network, the circle of people that lets the individual feel a sense of belonging to a part of a community [5]. Sensitivity to psychological distress and emotional loneliness is said to increase the risk of dementia; however, there is not as much evidence for this phenomenon as in the field of social loneliness. Conversely, an expansive social network might prevent cognitive decline in older age [6]. Additionally, accumulating evidence indicates that loneliness itself could cause a depletion in sensory and cognitive stimulation which results in a decrease in neural reserve [7,8]. Over the years, limited research has sought to establish a link between social isolation and loneliness and the onset of dementia, but the nature of this relationship remains relatively unexplored. Notwithstanding the increasing evidence linking loneliness to the risk of dementia, the published literature has shown a degree of inconsistency.

The question could arise whether loneliness causes dementia or dementia causes loneliness. There is an assumption that loneliness may be a behavioral reaction to diminished cognition. Moreover, impaired social cognition and diminished sociocognitive skills are typical for prodromal dementia [9]. The aim of this narrative review is to report on the recent literature and highlight the lines of research concerning potential relationships between loneliness and dementia.

## 2. Methodology

In this comprehensive narrative review, we directed our attention toward acquiring a holistic perspective on the subject matter concerning the relationship between loneliness and dementia. Our overarching hypothesis posited that feeling lonely exhibits correlations with cognitive impairments and the onset of dementia. To systematically collect pertinent scientific literature, we conducted a methodical search across various bibliographic databases, including PubMed, Google Scholar, and Web of Science. This search was guided by a predefined hypothesis and a set of inclusion and exclusion criteria. Only the most pertinent and current research findings were considered for inclusion in this review.

The primary search strategy involved the utilization of specific keywords, which encompassed the following: “loneliness” OR “feeling lonely” OR “being alone” AND “dementia” OR “Alzheimer’s Disease” OR “cognitive decline” AND “relation*”. The inclusion criteria were constrained to articles published in English between the years 1990 and 2023, encompassing clinical studies, in vivo and in vitro investigations, systematic reviews, and meta-analyses. The final selection of studies for review was made through a consensus-driven decision-making process by two authors, resulting in the inclusion of 112 resources. Articles were excluded if they did not align with the specific criteria outlined for inclusion or if they redundantly reiterated combinations of phrases.

## 3. Dementia and Its Risk Factors

Dementia is a disorder characterized by a decline in cognition that interferes with independent, daily functioning. It is caused by an interplay between genes, lifestyle, and environment. Many factors of this condition were revealed with the development of genetic research. There are different types of dementia such as dementia in Alzheimer’s disease (AD), frontotemporal lobar degeneration (FLD), Lewy body dementia (LBD), or vascular dementia (VaD). There are sex differences in the most common types of dementia—women are more often affected by AD while men suffer more from FTD, LBD, and VaD [10].

Risk factors of dementia are indisputably age and family history. Likewise, genetic susceptibility is a significant factor in the development of the disease, such as apolipoprotein Eε4 allele [11]. However, none of these risk factors can be modified significantly by changes in behavior or medical treatment. Hence, discovering new triggers of dementia that can be eliminated or diminished by human intervention would be a significant improvement in the treatment. An example of this could be diabetes, which has a positive correlation with dementia [12]. Similar observations have revealed the influence of midlife obesity and midlife hypertension on dementia as well [13]. A strong connection was also observed between smoking and increased risk for cognitive decline [14]. The protective effect of physical activity and cognitive training was proven in research [15]. The association between gastrointestinal tract microbiota and dementia has been established by several studies [16]. Finally, there is growing evidence that loneliness may contribute to dementia (Table 1).

## 4. Risk Factors and Protective Elements of Loneliness and Dementia

Dementia and loneliness exhibit a multitude of shared risk factors encompassing environmental, lifestyle, and genetic dimensions. 

There is evidence to suggest that genetics can play a role as a shared risk factor for both loneliness and dementia, although the mechanisms and genetic factors involved may differ. Research indicates a heritable component to loneliness, suggesting that genetic factors contribute to an individual’s predisposition to feeling lonely. Studies, including twin and family studies, have shown that loneliness has a heritability estimate, meaning that a portion of the variation in loneliness can be attributed to genetic factors [22]. Similarly, genetics plays a significant role in the risk of developing dementia, particularly in certain forms like Alzheimer’s disease. There are known genetic risk factors, including specific gene variants (e.g., APOE gene) that are associated with an increased likelihood of developing dementia. Having a family history of dementia is also recognized as a risk factor, suggesting a genetic influence [23,24]. While the specific genetic factors contributing to loneliness and dementia may not be identical, there could be some overlap, especially in the broader context of mental health and cognitive function. Genetic factors may influence personality traits, cognitive abilities, and mental health, all of which can contribute to both loneliness and dementia [25,26]. 

It is important to note that genetics is just one piece of the puzzle, and environmental, lifestyle, and other factors also play crucial roles in the development of both loneliness and dementia. Loneliness and dementia are intertwined with social isolation, a recognized environmental risk factor for loneliness. Individuals lacking social connections may experience cognitive decline due to fewer cognitive stimulations and diminished social support [27]. We delve into this subject in greater detail in the upcoming section. The social environment plays a crucial role in loneliness and dementia. Living in socially deprived areas or neighborhoods with limited community engagement can contribute to feelings of loneliness. Additionally, neighborhood characteristics, such as limited access to resources and social activities, may impact cognitive function [28,29]. Cultural norms, societal attitudes, and community values influence both loneliness and dementia. These factors shape an individual’s sense of belonging and social integration, while broader societal aspects like education, cultural practices, and attitudes toward aging impact cognitive health [30,31]. Healthcare access is pivotal for mental health in both loneliness and dementia. Limited access to healthcare resources may contribute to feelings of loneliness. Adequate healthcare access is crucial for the early detection and management of cognitive decline [32,33]. Economic factors are significant contributors to loneliness and dementia. Financial hardship can impact an individual’s ability to engage in social activities, leading to loneliness. Moreover, lower socioeconomic status is associated with an increased risk of dementia [34,35]. Life transitions and events play a role in both loneliness and dementia. Transitions such as retirement, loss of a spouse, or relocation can contribute to loneliness. Stressful life events may be involved in the development or progression of dementia, and the resulting loneliness could act as a contributing factor [36,37].

Individuals experiencing loneliness exhibit a higher propensity for engaging in behaviors such as smoking and a reduced likelihood of participating in regular exercise [38,39]. Both smoking and inadequate physical activity have been implicated in the risk of dementia [40]. Furthermore, loneliness correlates with clinical risk factors known to elevate the risk of cognitive impairment, including hypertension, obesity, and diabetes [41,42,43]. Loneliness is also associated with heightened physiological dysregulation in response to stress, such as an increased inflammatory response to acute stressors, a factor linked to dementia risk [44,45,46]. Depressive symptoms and negative affect play a role in the connection between loneliness and compromised health, mechanisms likely extending to cognitive aspects. For instance, negative affect has been identified as a mediator in the longitudinal relationship between loneliness and deteriorating health over time and negative affect serves as a risk factor for incident dementia [47,48]. We discuss this topic more comprehensively in Section 7.

Another aspect is that loneliness and dementia share some protective elements, which can both help mitigate loneliness and support individuals with dementia. Addressing loneliness in individuals with dementia involves fostering social engagement, utilizing technology for communication, and creating a supportive environment [49,50,51]. Meaningful activities, cognitive stimulation, and therapeutic interventions such as music and art therapy contribute to emotional well-being [52,53]. Caregiver education and support for them are crucial in reducing stigma and promoting understanding [54]. A comprehensive, personalized approach is essential to enhance the quality of life for those with dementia and alleviate loneliness.

## 5. Feeling Lonely, Being Alone, and Their Connection with Dementia

Loneliness is defined as the subjective experience of isolation and a deficit in feelings of social fulfillment [55]. It was suggested that after adjustment for other risk factors, older people with feelings of loneliness were more likely to develop dementia than people without such emotions [17]. Feeling lonely rather than being alone (defined as being not or no longer married, living alone, or having a small social network, little participation in activities with others, or lack of social engagement) may be associated with an increased risk of clinical dementia in later life [8]. The literature shows that people living alone are more isolated from their family members and experience more emotional loneliness than those who live with others; however, they do not have reduced contact with friends and engage with regular frequency in social activity. Two-years follow-ups revealed no association between living alone and poorer cognitive function [18]. Research by Zebhauser demonstrated that around 70% of people living alone expressed no feelings of loneliness [56]. However, unmarried individuals indicated feeling lonelier than married individuals [17,57]. 

Persistent loneliness could be a dangerous risk factor for neurocognitive impairment, whereas transient loneliness may be a protective factor for the development of dementia compared to no loneliness. This phenomenon may show differences depending on personality traits [19]. According to the WHO, in the comparison of three groups—people experiencing no loneliness, people who experience transient loneliness, and those exposed to persistent loneliness—patients from the second group have the lowest risk for developing dementia. Nevertheless, some studies conducted in China suggested that both chronic and transient loneliness may lead to cognitive decline, but the impact of persistent loneliness is more significant [58]. The association might be explained by differences in neurobiological mechanisms between those groups. 

Elucidating the linkage between loneliness and the risk of developing dementia, and determining whether it stems from social causation or social selection, represents a significant issue of scientific concern. There are two hypotheses—the social causation hypothesis states that loneliness itself increases the risk of dementia, whereas under the social selection hypothesis the risk of dementia among lonelier people is more likely to occur because of their greater underlying biological and environmental risk exposure. For instance, in pairs of monozygotic twins, differences in the effect of loneliness on dementia risk should be attributed to the twins’ dissimilarity in environmental exposure, as they share all of their genotypes and common environments [20]. 

Referring to the literature, previous studies have examined connections between sex, loneliness, and the risk of dementia.

Studies on the Chinese population revealed that the correlation between loneliness and dementia is stronger in the male than female gender [21].

The recent outbreak of the COVID-19 pandemic forced many changes in social life—social isolation and restrictions in contact even with family and close friends. People whose social contacts were usually limited now had to face an even greater reduction. In studies, it has been well documented that the pandemic had a negative influence on mental health, increasing the feeling of loneliness [59]. During these times, social media started to have a greater impact on social connections and elderly people who were not as familiar with modern technology as young adults were socially excluded [60]. The main way of contact during the pandemic was Internet communication—writing and making video calls. Based on the current evidence, it is difficult to say whether video calls help to reduce loneliness in older people [61].

## 6. Sociocognitive Skills and Neurobiology

One of the most crucial factors that might account for cognitive development is social interaction. There are two theories supporting this thesis. The first one is called the “use it or lose it” theory in which engaging in social or cognitive activities stimulates the brain and the lack of them results in brain atrophy [62]. The cognitive reserve theory indicates that neurogenesis and the synaptic density of the brain may be increased by social connections. As a result, the cognitive reserve of an individual is greater and therefore the risk of neurodegeneration, including dementia, could be lower [9,63]. Further studies confirm that the quality and quantity of sociocognitive skills used in social interactions are closely related to the level of loneliness and prevention of dementia [7]. It is possible that being alone without cognitive stimulation leads to a decrease in neural reserve and therefore to impaired social cognition. Prodromal dementia is characterized by diminished sociocognitive skills and could result in loneliness, which creates a correlation phenomenon between loneliness and dementia [49,64]. 

The influence of loneliness on dementia may be explained by neurobiological changes. Loneliness could compromise neural systems responsible for cognition and memory, hence making individuals more susceptible to neurodegeneration. Alterations in the brain are the effects of the underexpression of genes that are downregulated in Alzheimer’s disease and genes taking part in mitochondrial dysfunction as well as in oxidative phosphorylation [27]. The affected regions include the orbitofrontal cortex and the medial prefrontal cortex which stimulate the hypothalamic–pituitary–adrenocortical axis (HPA). The HPA is responsible for increasing the level of cortisol activation after stress triggers [46,65]. Due to long-term loneliness, persistent stress especially activates the sympathetic nervous system which may upregulate inflammatory gene expression, downregulate antiviral responses, and cause greater activation of glucocorticoids and hypercortisolism, stimulating the HPA in turn [66,67,68,69]. The dysregulation of the HPA results in the destruction of the hippocampus—one of the most important structures in the development of dementia and AD [7,70]. Other brain structures changed by loneliness are gray matter volume in the left posterior superior temporal sulcus and decreased regional white matter density in the bilateral inferior parietal lobe, right anterior insula, and posterior temporoparietal junction [71]. Activities in the dorsomedial prefrontal cortex, posterior cingulate cortex, and posterior insular cortex are also observed to be diminished among lonely individuals [72,73]. It is of interest to know that all these regions are taking part in memory, attention control, executive function, and social brain functions. Another argument for the molecular relationship between loneliness and dementia may be evidence from certain experiments on animals. Previous research showed that Alzheimer’s disease was associated with social isolation through the diminished neurotrophic factor and higher oxidative stress in brain neurons [74]. Finally, high levels of pathologic deposits of the peptide amyloid β and the protein tau typical for Alzheimer’s disease were found in the right entorhinal cortex and right fusiform gyrus among people who experience loneliness (Figure 1) [75]. 

Certain aspects of the neurobiological evidence for loneliness causing dementia remain unclear, and the research differs in its results in this context [7]. 

## 7. Visual and Hearing Impairment

The loss of sensory abilities with age is a natural process, yet it results in detrimental effects on both physical and psychological conditions, including social activity which is a preventing factor for dementia. The main problems are visual and hearing impairment as essential senses for communication. 

It was observed that 43.5% of people over 60 years old with visual impairment suffer from moderate to severe loneliness [77]. Hearing impairment affecting over 40% of people over 60 years old is observed to be a risk factor for dementia [78]. It is possible that the loss of hearing may be causally associated with dementia through social isolation [79]. Loneliness in elderly people might also mediate the relationship between sensory loss, including hearing and visual impairment, and psychiatric symptoms such as delusions, hallucinations, depressive symptoms, and anxiety which aggravate social isolation [80,81,82]. 

However, research results differ on the influence of sensory loss on dementia mediated by loneliness [83,84].

## 8. Depression and Psychosis

Multiple studies propose a connection between depressive and psychotic symptoms in elderly individuals with dementia and loneliness within this demographic. According to Shen et al., there is a positive correlation between dementia and loneliness, and in 75% of such cases depressive symptoms were also occurring [27].

In the course of dementia, behavioral and psychological symptoms in dementia (BPSD) including agitation, aberrant motor behavior, anxiety, elation, irritability, depression, apathy, disinhibition, delusions, hallucinations, and sleep or appetite changes may be present [85]. Research shows that elderly people with high emotional loneliness experience depressive symptoms and hallucinations [80,83,86,87,88]. The correlation with the latter seems to occur among individuals with Alzheimer’s disease or dementia as well; however, some studies deny it [89]. 

For a considerable period, the association between depression and memory dysfunction, including dementia, has been recognized, but its precise nature remains incompletely understood. Depression may be both a risk factor and a prodromal sign of ongoing dementia, yet more research inclines toward the first aspect. People with depression are said to have a higher risk of vascular dementia, mild cognitive impairment, and Alzheimer’s disease [90]. It is estimated that for individuals suffering from recurring depression, each episode is associated with a 13% higher rate of dementia [91]. The greatest risk of dementia can be associated with both midlife and late-life depression [92]. The explanation for depression being a prodrome condition of dementia might be the glucocorticoid hypothesis. Stress in the course of depression causes the overproduction of glucocorticoids which in turn overactivates hippocampal receptors and consequently leads to the atrophy of this brain structure associated with dementia (Figure 2) [93,94,95]. What is more, in a lot of studies, depression occurs closely in time with dementia among the elderly. It could be assumed that late-onset depression may be a prodromal sign of neurostructural changes in dementia [96]. Cognitive impairment in depression often does not subside—dementia in the dementia syndrome of depression (DS) is more frequent in this group than among individuals with depression alone [97]. Nevertheless, there are studies that appear to be inconsistent with the conception of a correlation between depression and dementia [98]. 

It is worth noting that the impact of depression in the older population can be assessed using the Geriatric Depression Scale (GDS) [99]. One key aspect is the inclusion of inquiries not only about loneliness but also about instrumental activities of daily living (IADLs). IADLs are activities that are necessary for living independently in the community. These activities go beyond basic self-care tasks (activities of daily living or ADLs) and include tasks such as managing finances, preparing meals, shopping, and using transportation [100]. Financial capacity is specifically highlighted as a critical aspect in the assessment. This is significant because financial challenges can have a substantial impact on the daily lives of older adults [101]. Issues related to financial management, such as budgeting, paying bills, and handling financial transactions, can contribute to stress and anxiety, potentially exacerbating or being exacerbated by depressive symptoms. In the research conducted by Giannouli et al., it was demonstrated that cognitive impairment and depression not only coexist but also contribute to a decline in financial capacity among patients [102]. This highlights a novel examination approach for clinicians when dealing with this specific population [102]. It is also underlined that a thorough examination is essential to determine whether legal guardianship is warranted [103]. The inclusion of questions related to IADLs, and specifically financial capacity, in the GDS recognizes that depression in older adults is not limited to emotional and psychological symptoms but also encompasses practical and functional aspects of daily life. 

## 9. Reducing Loneliness among the Elderly

The COVID-19 pandemic forced generalized and rapid actions to reduce loneliness, especially among the elderly. Prior to the pandemic, this problem was widely addressed, and proper actions were implemented as well. Meeting centers that were opened in many countries were shown to have a positive impact on people with dementia and their caregivers, especially in terms of the feeling of burden and isolation [104]. An integrative review of such interventions concluded that active engagement in planning and organizing group activities was effective in tackling loneliness [105]. 

Besides face-to-face activities, methods involving digital technology were widely assessed. Certain studies have proved that video communication and telephone calls with family helped the elderly with social isolation [106]. However, there is data that contradicts these results [61]. Newly developed phone apps provide many possibilities for their users, such as meditation or mindfulness programs, conversational agents, or even matching individuals for maintaining regular online contact [107]. There are results showing that chatbots were perceived positively and were helpful in reducing loneliness for the majority of older adults [108]. It is worth noting that prior to the pandemic, 75% of individuals aged 65 years and older used digital technology as a routine aspect of their daily lives according to the Stanford Center on Longevity.

## 10. Final Remarks and Conclusions

Dementia undeniably persists as a significant medical, social, and economic challenge in the 21st century. Concurrently, loneliness poses a noteworthy concern in contemporary society [1,2]. Both of them are being fought against with the use of many different methods, but none of these appear to be especially effective [51]. An increasing number of studies indicate that a correlation between loneliness and dementia may exist. An important issue in the context of proper understanding of this phenomenon is to distinguish between feeling lonely and being alone [109]. It is suggested that the more probable risk factor for dementia in this context is the first one, but studies investigating individuals living alone are ambiguous [8,110]. Accumulating evidence indicates that enduring loneliness may serve as a risk factor for cognitive decline, while transient loneliness appears to provide a protective effect [19]. Various hypotheses have been proposed to account for the propensity towards dementia in the presence of loneliness, emphasizing either the direct influence of social isolation (social causation hypothesis) or individual vulnerabilities predisposing to this association (social selection hypothesis) [111]. Nevertheless, the results of additional investigation have inclined towards the latter hypothesis. The effect of the environment on both the quality and quantity of social interactions could be a crucial aspect in developing or preventing dementia [62]. Interpersonal interactions are said to facilitate the development of sociocognitive abilities, which serve as a mechanism for managing relationships and thereby promote cognitive capacity [7]. However, not only loneliness but also prodromal dementia co-occurs with sociocognitive skills [49,64,112]. A similar phenomenon appears when the issue of BPSD is considered depending on an individual’s behavior [85]. Sensory loss, especially hearing impairment, may be an essential obstacle in relationships as well. Research indicates that it could lead to both loneliness and dementia [77,78]. Finally, being or feeling lonely might structurally change many regions of the brain. Orbitofrontal and medial prefrontal cortex, when affected, overstimulate the HPA, which has a detrimental influence on the hippocampus—a structure crucial for memory [7,46,65]. Nonetheless, it is worth noting that the bidirectional relationship between loneliness and dementia can be possibly explained by their shared additional biological or environmental risk or preventive factors [14,22,23,24,25,26,27,28,29,30,31,32,33,34,35,36,49,50,51,52,53,54].

There are many studies researching and still discovering new aspects of the relationship between loneliness and dementia. The question of whether loneliness and which types of it act on dementia, or conversely, remains unsolved. There is a need for more studies that properly differentiate between feeling alone and social isolation, as well as precisely select research groups regarding the type of dementia, comorbidity, and environmental aspects. Neurobiological and molecular analysis is also required for a better understanding of the complex mechanisms between loneliness and dementia. Objective results would contribute to improving both the diagnosis and treatment of neurodegenerative diseases in the future. The ultimate goal is to stimulate constructive and collective thinking for more effective personalized interventions [3]. The importance of understanding the identified causal links was emphasized for better comprehension of these psychosocial concepts. However, this review acknowledges the need for further research to reduce or eliminate undesired effects.

## Figures and Tables

**Figure 1 ijms-25-00271-f001:**
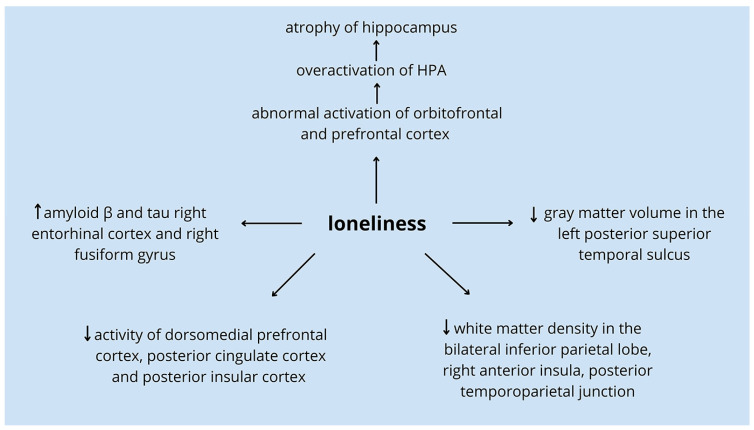
Loneliness affects different regions of the brain responsible for higher cognitive functions [46,65,66,67,68,69,71,72,73,74,75,76]. Loneliness may affect the activation of orbitofrontal and prefrontal cortex leading to overactivation of HPA and, ultimately, atrophy of hippocampus—one of the most important structures in the development of dementia and AD [7,46,65,66,67,68,69,70]. Other brain structures changed by loneliness are gray matter volume in the left posterior superior temporal sulcus and decreased regional white matter density in the bilateral inferior parietal lobe, right anterior insula, and posterior temporoparietal junction [71]. Activities in the dorsomedial prefrontal cortex, posterior cingulate cortex, and posterior insular cortex are also observed to be diminished among lonely individuals [72,73]. It is of interest to know that all these regions are taking part in memory, attention control, executive function, and social brain functions. Finally, high levels of pathologic deposits of the peptide amyloid β and the protein tau typical for Alzheimer’s disease were found in the right entorhinal cortex and right fusiform gyrus among people who experience loneliness [75]. Legend: ↑—higher level; ↓—lower level; longer arrow—loneliness results in.

**Figure 2 ijms-25-00271-f002:**
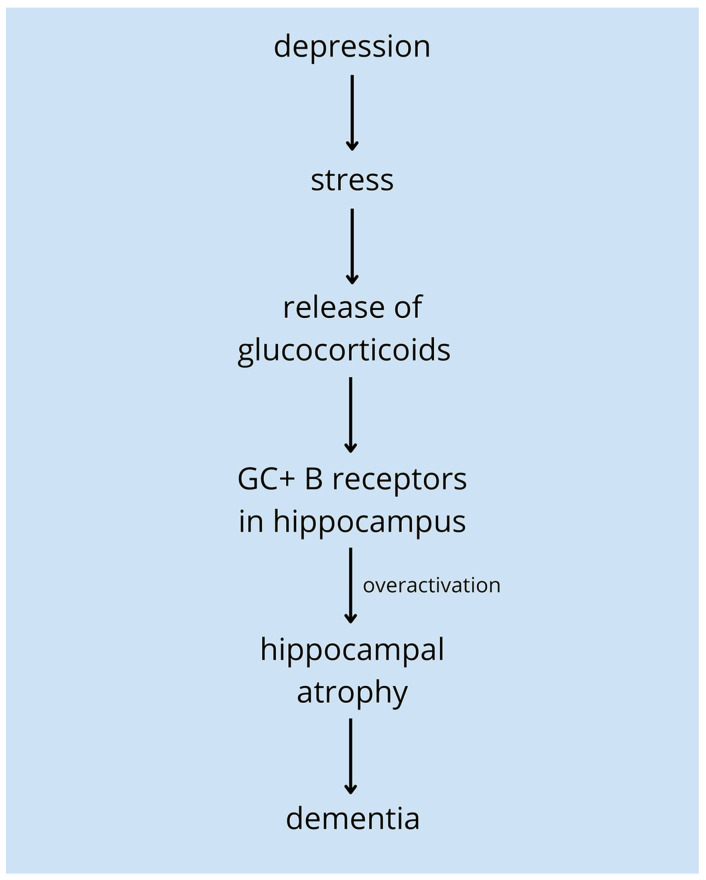
The glucocorticoid hypothesis. In the course of depression, stress triggers the release of glucocorticoids from the adrenal glands. These hormones act on their B-receptors (GC+ B receptors) in the hippocampus, overactivating them. Consequently, hippocampal atrophy occurs, which may lead to dementia [93].

**Table 1 ijms-25-00271-t001:** Main studies concerning the relationship between loneliness and dementia.

Author, Year, Citation	Type of Study	Group	Main Findings
S. B. Rafnsson et al., 2020, [17]	Longitudinal, cohort study	6677 healthy individuals	Loneliness was positively correlated with dementia after more than 6 years, and negatively to being married.
T. J. Holwerda et al., 2014, [8]	Longitudinal, cohort study	2173 people without dementia	Older people with feelings of loneliness were more likely to develop dementia after 3 years.
I. E. M. Evans et al., 2019, [18]	Longitudinal study	2197 people without dementia	No association between living alone and poorer cognitive function at 2-years follow-up.
S. C. Akhter-Khan et al., 2021, [19]	Longitudinal, cohort study	2880 participants	Persistent loneliness is a risk factor for neurocognitive impairment, whereas transient loneliness may be a protective factor.
A. J. Kim et al., 2021, [20]	Longitudinal study	2476 individual twins	Mild levels of isolation are related to greater dementia risk; however, there is a greater environmental impact.
Z. Zhou et al., 2018, [21]	Longitudinal, cohort study	7867 healthy individuals	There is a greater correlation between isolation and dementia among men.

## Data Availability

No new data were created or analyzed in this study. Data sharing is not applicable to this article.

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
