# Peer review of "Correlations between Dementia and Loneliness"

_ijms, 2023, doi:10.3390/ijms25010271_

Round 1

Reviewer 1 Report

Comments and Suggestions for Authors

Dear Authors,

Thank you for your interesting study. The importance of both diseases in contemporary medicine and society, as well as their mutual relation and sometimes  even difficulties in distinguishing them, clearly demand necessity for further investigation. Your research of 73 studies from the period from1990. to 2023. contributes greatly to better understanding of this particular problem. Using the right methodology, you succeeded in explaining  the differences, in pointing out the risk factors and suggesting the way how to reduce loneliness in elderly people. Two figures and the list of 72 references add to increased quality of you work. I would like to suggest to continue further on.

Author Response

We thank the Reviewer for encouraging feedback and appreciate the insightful comments and suggestions.

We expanded our review of recently added sections, encompassing the key subjects within the realms of dementia and loneliness, such as the common risk and preventive factors. Additionally, we incorporated fresh references, bringing the total count to 115.

We hope that the introduced revisions significantly improve the quality of this review and qualify it for further editorial stages.

Sincerely,

Authors

Reviewer 2 Report

Comments and Suggestions for Authors

Major comments

1. The introduction leaves the reader uncertain about the rationale for the proposed hypothesis: “Our overarching hypothesis posited that feeling lonely may exhibit associations with cognitive impairments and the onset of dementia. ”

2. While the paper reviews evidence supporting the hypothesis that “feeling lonely may exhibit associations with cognitive impairments and the onset of dementia”, it did not deliver a conclusion. Greater emphasis on distinguishing correlation from causation is necessary. For example, the authors wrote “It is of interest to note that these disturbances can be caused by loneliness, or they could lead to it, creating regenerative feedback. Research shows that elderly people with high emotional loneliness experience depressive symptoms and hallucinations [47, 50, 53–55].” First of all, the pronouns in “or they could lead to it” are unclear.  While the paper suggests a bidirectional relationship between loneliness and certain psychological disturbances, it presents no causative proof. It is plausible that the reason for this connection is that loneliness and those two symptoms are both triggered by additional biological or environmental risk factors. 

3. As a reader, I’d be interested if the authors could have a discussion on common risk factors or protective elements shared by loneliness and dementia, such as genetic, environmental, and lifestyle influences.

Minor comments:

1. The review occasionally mentions data or findings without proper referencing, for example, epidemiological data and links between loneliness and dementia. Citations are crucial for these assertions. For example, in the Introduction section, the authors stated many data or previous findings without citations, including epidemiologic findings line 27-31) and the association between loneliness and dementia (line 32, line 38-39).

2. The phrasing needs refinement for precision. Rather than suggesting "dementia risk among lonelier people is more likely to occur," it should be reworded to either "dementia …. is more likely to occur" or "the risk of dementia … is higher."

3. Please revise the phrase in this current draft to be accurate. For instance, the statement of “dementia risk among lonelier people is more likely to occur…” The author should phrase it as “dementia … is more likely to occur” or “dementia risk …. is higher”.

4. “According to Shen et al., the 204 fully adjusted hazard ratio for dementia related to loneliness is 1.04 with 75% of this relationship being attributed to depressive symptoms” While it is important to state the statistics, I would suggest the authors to phrase it in a more layman-friendly way.

5. “Both loneliness and dementia are undoubtedly two great medical, social, and economic problems of the XXI century.” Describing loneliness as a significant medical and economic challenge alongside dementia might overstate its impact. It would be more accurate to contextualize loneliness as a contributing factor to these broader issues without equating its severity with that of dementia.

Comments on the Quality of English Language

Please see my minor comments above.

Author Response

We thank the Reviewer for encouraging feedback and appreciate the insightful comments and suggestions.

Below, we provide a point-by-point response to each of the reviewer’s comments. 

We hope that the introduced revisions significantly improve the quality of this review and qualify it for further editorial stages. 

Sincerely,

Authors

Major comments:

  1. The introduction leaves the reader uncertain about the rationale for the proposed hypothesis: “Our overarching hypothesis posited that feeling lonely may exhibit associations with cognitive impairments and the onset of dementia. ”

Response: Thank you for this valuable suggestion. We added new aspects of correlations between loneliness and dementia to the introduction with proper references to provide the rationale.

2. While the paper reviews evidence supporting the hypothesis that “feeling lonely may exhibit associations with cognitive impairments and the onset of dementia”, it did not deliver a conclusion. Greater emphasis on distinguishing correlation from causation is necessary. For example, the authors wrote “It is of interest to note that these disturbances can be caused by loneliness, or they could lead to it, creating regenerative feedback. Research shows that elderly people with high emotional loneliness experience depressive symptoms and hallucinations [47, 50, 53–55].” First of all, the pronouns in “or they could lead to it” are unclear.  While the paper suggests a bidirectional relationship between loneliness and certain psychological disturbances, it presents no causative proof. It is plausible that the reason for this connection is that loneliness and those two symptoms are both triggered by additional biological or environmental risk factors. 

Response: Thank you very much for this valuable comment. We emphasized the probable character of the loneliness and dementia relationship as a correlation. We deleted the sentence ,,It is of interest to note that these disturbances can be caused by loneliness, or they could lead to it, creating regenerative feedback". We highlighted the assumption that the possible reason for correlations between  loneliness and dementia is that they may be triggered by additional biological or environmental risk factors. 

3. As a reader, I’d be interested if the authors could have a discussion on common risk factors or protective elements shared by loneliness and dementia, such as genetic, environmental, and lifestyle influences.

Response: Thank you for this valuable suggestion. We added the whole chapter ,,4.  Risk factors and protective elements of loneliness and dementia".

Minor comments:

  1. The review occasionally mentions data or findings without proper referencing, for example, epidemiological data and links between loneliness and dementia. Citations are crucial for these assertions. For example, in the Introduction section, the authors stated many data or previous findings without citations, including epidemiologic findings line 27-31) and the association between loneliness and dementia (line 32, line 38-39).

Response: Thank you for your comment. We added proper references to the mentioned epidemiological data (reference 1 and 2).

2. The phrasing needs refinement for precision. Rather than suggesting "dementia risk among lonelier people is more likely to occur," it should be reworded to either "dementia …. is more likely to occur" or "the risk of dementia … is higher."

Response: Thank you for this important suggestion. We corrected all of the mentioned mistaken phrases.

3. Please revise the phrase in this current draft to be accurate. For instance, the statement of “dementia risk among lonelier people is more likely to occur…” The author should phrase it as “dementia … is more likely to occur” or “dementia risk …. is higher”.

Response: Thank you for your comment. We changed the sentence into: ,,

There are two hypotheses - the social causation hypothesis which states that loneliness itself increases the risk of dementia, whereas under the social selection hypothesis the risk of dementia among lonelier people is more likely to occur because of their greater underlying biological and environmental risk exposure".   4. “According to Shen et al., the 204 fully adjusted hazard ratio for dementia related to loneliness is 1.04 with 75% of this relationship being attributed to depressive symptoms” While it is important to state the statistics, I would suggest the authors to phrase it in a more layman-friendly way.   Response: Thank you for this valuable suggestion. We changed this sentence into: ,,

According to Shen et al., there is a positive correlation between dementia and loneliness, what is more in 75% of such cases depressive symptoms were also occurring [29]".

5. “Both loneliness and dementia are undoubtedly two great medical, social, and economic problems of the XXI century.” Describing loneliness as a significant medical and economic challenge alongside dementia might overstate its impact. It would be more accurate to contextualize loneliness as a contributing factor to these broader issues without equating its severity with that of dementia.

Response: Thank you for this suggestion. We changed the description into: ,,Dementia undeniably persists as a significant medical, social, and economic challenge in the 21st century. Concurrently, loneliness poses a noteworthy concern in contemporary society [1,2]"

Reviewer 3 Report

Comments and Suggestions for Authors This is a debate topic and it is of interest to the readers, but a major revision is needed. The number of references needs to be larger in order to support the Introduction as well as the Discussion. A very relevant review that is relevant and the authors should discuss: Tragantzopoulou, P., & Giannouli, V. (2021). Social isolation and loneliness in old age: Exploring their role in mental and physical health. Psychiatriki, 32(1), 59-66. In addition Figure 1 should be explained in a separate paragraph. There is a sentence in a language other than English in page 6. Finally, authors should consider adding the influence of depression as measured with Geriatric Depression Scale (GDS) which contains a question over loneliness and IADLs such as financial capacity which is of extreme importance for the everyday life of older adults. There is a plethora of recent relevant research. This could be an additional section with great implications for different types of dementias-neurocognitive disorders, for which authors can read at https://scholar.google.com/scholar?hl=el&as_sdt=0%2C5&q=financial+capacity+depression+giannouli&btnG=

Comments on the Quality of English Language

 Moderate English language editing is needed.

Author Response

We thank the Reviewer for encouraging feedback and appreciate the insightful comments and suggestions.

Below, we provide a point-by-point response to each of the reviewer’s comments. 

We hope that the introduced revisions significantly improve the quality of this review and qualify it for further editorial stages. 

Sincerely,

Authors

1. The number of references needs to be larger in order to support the Introduction as well as the Discussion.

Response: Thank you for this valuable suggestion. We incorporated fresh references into both introduction and discussion, bringing the total count of references to 115.

2. A very relevant review that is relevant and the authors should discuss: Tragantzopoulou, P., & Giannouli, V. (2021). Social isolation and loneliness in old age: Exploring their role in mental and physical health. Psychiatriki, 32(1), 59-66.

Response: Thank you for this comment. We described the relevant aspects of the mentioned article with its citation in the introduction.

3. In addition Figure 1 should be explained in a separate paragraph.

Response: Thank you for this important suggestion. We added a paragraph describing Figure 1 above the figure.

4. There is a sentence in a language other than English in page 6. 

Response: Thank you for this comment. We deleted this sentence.

5. Finally, authors should consider adding the influence of depression as measured with Geriatric Depression Scale (GDS) which contains a question over loneliness and IADLs such as financial capacity which is of extreme importance for the everyday life of older adults.

Response: Thank you for this valuable comment. We added a whole paragraph concerning this issue in the end of chapter 8.

6. There is a plethora of recent relevant research. This could be an additional section with great implications for different types of dementias-neurocognitive disorders, for which authors can read at https://scholar.google.com/scholar?hl=el&as_sdt=0%2C5&q=financial+capacity+depression+giannouli&btnG=

Response: Thank you for this crucial suggestion. We added a new paragraph concerning this issue in chapter 8.